# Abiotic Stress-Induced Leaf Senescence: Regulatory Mechanisms and Application

**DOI:** 10.3390/ijms241511996

**Published:** 2023-07-26

**Authors:** Shuya Tan, Yueqi Sha, Liwei Sun, Zhonghai Li

**Affiliations:** State Key Laboratory of Tree Genetics and Breeding, College of Biological Sciences and Technology, Beijing Forestry University, Beijing 100083, China

**Keywords:** leaf senescence, abiotic stress, stress tolerance, transcription factor, *Arabidopsis*, crop

## Abstract

Leaf senescence is a natural phenomenon that occurs during the aging process of plants and is influenced by various internal and external factors. These factors encompass plant hormones, as well as environmental pressures such as inadequate nutrients, drought, darkness, high salinity, and extreme temperatures. Abiotic stresses accelerate leaf senescence, resulting in reduced photosynthetic efficiency, yield, and quality. Gaining a comprehensive understanding of the molecular mechanisms underlying leaf senescence in response to abiotic stresses is imperative to enhance the resilience and productivity of crops in unfavorable environments. In recent years, substantial advancements have been made in the study of leaf senescence, particularly regarding the identification of pivotal genes and transcription factors involved in this process. Nevertheless, challenges remain, including the necessity for further exploration of the intricate regulatory network governing leaf senescence and the development of effective strategies for manipulating genes in crops. This manuscript provides an overview of the molecular mechanisms that trigger leaf senescence under abiotic stresses, along with strategies to enhance stress tolerance and improve crop yield and quality by delaying leaf senescence. Furthermore, this review also highlighted the challenges associated with leaf senescence research and proposes potential solutions.

## 1. Introduction

The leaves of plants serve as the primary sites for photosynthesis, where light energy is converted into chemical energy stored in carbohydrate molecules. These carbohydrates serve as the main energy source for all living organisms on Earth. Senescence, the final stage of leaf development, is a gradual and intricate biological process comprising initiation, progression, and terminal phases [1,2]. In this process, the leaves gradually turn yellow, shrivel, and fall off. During the later stages of leaf senescence, chlorophyll and chloroplasts deteriorate, accompanied by the breakdown of macro-molecules like proteins, lipids, and nucleic acids [1,2]. In annual plants, the nutrients released from senescent leaves are transferred to actively growing young leaves and seeds to enhance reproductive success. In the case of perennial plants, such as deciduous trees, nitrogen from leaf proteins is redirected to form bark storage proteins in phloem tissues. These proteins are stored throughout the winter and then mobilized and reused for spring shoot growth [3,4,5]. In agriculture, senescence is capable of remobilizing leaf nitrogen and micronutrients into the grain or fruit. The NAC transcription factor NAM-B1 plays an important role in the regulation of expressions of nitrogen transport-related genes during senescence [6]. A recent study revealed that OsDREB1C shortens lifespan but improves photosynthetic capacity and nitrogen utilization, and transgenic plants with overexpression of OsDREB1C have 41.3% to 68.3% higher yields than wild-type plants [7]. Consequently, the timing of leaf senescence plays a crucial role in facilitating nutrient cycling, environmental adaptation, and reproduction in plants [8].

The leaf senescence process is accompanied by changes in the expression of thousands of senescence-associated genes (*SAG*s) [9]. Studies have shown that several transcriptional regulators (TFs) regulate senescence by controlling *SAG* expression [10]. In one of them, a number of NAC TFs were identified as core regulators of senescence [11,12,13]. EIN3, a key TF that functions downstream of EIN2 in ethylene signaling pathway, increases the transcript levels of *ORE1/AtNAC092/AtNAC2* through the direct repression of *miR164* transcription [14]. WRKY53 positively regulates leaf senescence [15] via targeting various *SAG*s such as *SENRK1* [16].

The initiation and progression of leaf senescence are influenced by various internal and external factors [1,2,8]. Leaf senescence can be triggered as a defense mechanism in response to biotic stress factors such as pathogen infection or insect damage. Additionally, abiotic stress factors including drought, high salinity, high temperature, or nutrient deficiencies can accelerate leaf senescence [1,2,8,17,18,19]. These stressors can induce oxidative stress, leading to the accumulation of reactive oxygen species (ROS), which can cause DNA damage and activate SAGs [1,2,8,17,18,20]. 

Numerous studies conducted on crops like wheat and rice demonstrated that modifying leaf senescence processes can have a significant impact on crop yield and quality. For instance, in apple trees (*Malus domestica*), improving fruit quality was achieved by extending the lifespan of leaves through the modulation of senescence-associated transcription factors, *MbNAC25* and *MdbHLH3* [21,22]. Similarly, in tomato (*Solanum lycopersicon*), increasing fruit yield and sugar content was achieved by suppressing the expressions of *SIORESARA1* (*ORE1*) and *SlNAP*, which delayed leaf senescence [23,24]. Additionally, delaying leaf senescence in tobacco or cassava resulted in enhanced drought resistance [25,26,27,28]. Therefore, gaining a deeper understanding of the regulatory mechanisms underlying leaf senescence can aid researchers in developing more resilient plants that can withstand environmental stresses. This, in turn, would lead to improvements in crop yield, quality, and contribute to global food security and sustainability [8]. This manuscript provides a comprehensive review of the molecular mechanisms involved in leaf senescence induced by abiotic stresses such as nitrogen deficiency, drought, high salinity, and extreme temperature. It also discusses strategies to enhance stress tolerance, crop yield, and quality by delaying leaf senescence. Furthermore, the review highlights the challenges associated with leaf senescence research and explores potential solutions.

## 2. Abiotic Stress-Induced Leaf Senescence

### 2.1. Nitrogen Deficiency-Induced Leaf Senescence

Nitrogen, an essential macronutrient in plants, plays a crucial role in leaf senescence, and its deficiency triggers a rapid senescence process [29,30,31]. ORE1, a key regulator of leaf senescence, was identified as a major factor in nitrogen deficiency-induced leaf senescence [29,30]. In conditions of nitrogen deficiency, loss of ORE1 function results in delayed senescence, while overexpression of ORE1 accelerates leaf senescence, characterized by yellowing leaves, reduced chlorophyll content, and increased expression of SAG12 [29]. Interestingly, overexpression of nitrogen limitation adaptation (NLA) in ORE1 overexpressing plants mitigates the leaf senescence phenotype induced by nitrogen deficiency. NLA, which encodes a RING-type ubiquitin ligase [32], represses leaf senescence by promoting the ubiquitination and degradation of the nitrate transporter NRT1.7 [33]. In a similar mechanism, NLA interacts with ORE1 in the nucleus and regulates its stability through polyubiquitination, with the involvement of PHOSPHATE2 (PHO2). PHO2 encodes an E2 ubiquitin-conjugating enzyme (UBC) and is responsible for maintaining cellular phosphate homeostasis in Arabidopsis [34,35]. Consequently, nla and pho2 mutant plants exhibit accelerated leaf senescence under nitrogen-starvation conditions, whereas nla/ore1 and pho2/ore1 double mutant plants retain green leaves. These findings suggest that fine-tuning the levels of ORE1 through post-translational modifications by NLA/PHO2 ensures a regulated progression of senescence [29]. Interestingly, the deubiquitinases UBP12 and UBP13 were identified as regulators of ORE1 stability by deubiquitinating polyubiquitinated ORE1 and increasing its stability [30]. Plants overexpressing UBP12 or UBP13 display accelerated leaf senescence, which can be reversed by mutation of ORE1. Conversely, overexpression of ORE1 exacerbates the senescence phenotype when UBP12 or UBP13 is also overexpressed [30]. These studies provided a model that explains the molecular framework underlying the involvement of ORE1 in the regulation of nitrogen deficiency-induced leaf senescence [29,30]. Under normal conditions, ORE1 is polyubiquitinated by the E3/E2 enzyme complex, NLA/PHO2, and, subsequently, degraded by 26S proteasomes, leading to delayed leaf senescence. However, under nitrogen-deficient conditions, UBP12 and UBP13 counteract the effects of NLA/PHO2 by deubiquitinating polyubiquitinated ORE1, preventing its degradation. This elevated level of ORE1 activates the expression of downstream SAG genes, thereby accelerating leaf senescence.

Recently, a zinc finger transcription factor called growth, development, and splicing 1 (GDS1) [36] was discovered to have a role in repressing leaf senescence induced by nitrogen deficiency [31]. GDS1 functions as a crucial co-activator or co-protein in the early stages of pre-mRNA splicing and is essential for growth and development in Arabidopsis [36]. Mutants of gds1 exhibit early leaf senescence, reduced NO_3_− content, and impaired nitrogen uptake under nitrogen-deficient conditions. Biochemical analysis revealed that GDS1 can bind to the G-box motifs present in the promoter regions of phytochrome-interacting factor 4 (PIF4) and PIF5, thereby repressing their expression [31]. PIF4 and PIF5 were identified as regulators of dark- and heat-induced as well as age-triggered leaf senescence in Arabidopsis [37,38,39,40]. Intriguingly, PIF4 and PIF5 also play a role in nitrogen deficiency-induced leaf senescence. Under nitrogen-deficient conditions, delayed leaf senescence was observed in pif4-2 and pif5-3 mutants compared to wild-type plants, while transgenic lines exhibited accelerated leaf senescence phenotypes. Expression levels of PIF4 and PIF5 in the leaves of wild-type plants were significantly higher under low nitrogen conditions compared to high nitrogen conditions [31]. This research presents a novel model to explain leaf senescence induced by low nitrogen levels [31]. Under nitrogen-sufficient conditions, GDS1 binds to the promoters of PIF4 and PIF5, inhibiting their expression and thereby suppressing the expression of downstream SAGs, resulting in delayed leaf senescence. However, under nitrogen-deficient conditions, the accumulation of anaphase-promoting complex or cyclosome proteins promotes the ubiquitination-mediated degradation of GDS1, leading to the release of PIF4 and PIF5 repression. Consequently, downstream SAGs are activated, promoting early leaf senescence.

Regarding both of these proposed models, which explain leaf senescence induced by low nitrogen levels [29,30,31]: are they independent or do they have any relationship? It was discovered that PIF4 and PIF5 directly bind to the promoter of *ORE1*, promoting its expression, thereby accelerating leaf senescence. Conversely, GDS1 directly binds to *PIF4* and *PIF5*, repressing their gene expression and mitigating low nitrogen-induced leaf senescence [31]. Future investigations will need to analyze whether GDS1 can directly regulate *ORE1* by binding to its promoters or indirectly influence its expression through PIF4/PIF5. Additionally, it would be interesting to explore if PUB12/14 and NLA/PHO2 can interact with GDS1. Furthermore, the relationship between these two regulatory pathways can be elucidated by generating multiple mutant combinations. These studies will contribute to a deeper understanding of leaf senescence induced by low nitrogen levels.

### 2.2. Drought Stress-Induced Leaf Senescence

Drought stress is a significant abiotic stress factor that has detrimental effects on plant growth and development [41], ultimately leading to leaf senescence [42]. The involvement of a NAC transcription factor, NTL4, in drought-induced leaf senescence has been identified [43]. Under normal conditions, there was no notable difference in the leaf senescence process between wild type plants, transgenic plants overexpressing *NTL4*, and *ntl4* mutants. However, under drought conditions, leaf senescence was accelerated in the transgenic plants while being significantly delayed in the *ntl4* mutant. NTL4 promotes the production of ROS by binding to the promoters of RBOHC and RBOHE under drought conditions. In turn, the elevated ROS production further stimulates *NTL4* gene expression, creating a feed-forward acceleration loop. Notably, *NTL4* is expressed at basal levels during vegetative growth stages and is rapidly induced in response to drought stress. The induction of *NTL4* expression under drought conditions is particularly evident in the distal leaf area, where leaf senescence initiates upon exposure to drought stress [2]. In response to drought, the distal regions of senescing leaves accumulate ROS and experience cell death [18]. This response facilitates the transfer of nutrients and metabolites from senescing leaves to absorptive organs and newly formed leaves, while minimizing water loss through transpiration [18]. Thus, NTL4-mediated leaf senescence enhances the chances of plant survival under drought conditions. Supporting this hypothesis, the overexpression of NAC transcription factors *ANAC019*, *ANAC055*, and *ANAC072* leads to early leaf senescence but increases drought tolerance [44]. Additionally, it has been found that the ABA receptor PYL9 promotes leaf senescence and enhances drought resistance [45]. By activating the signaling cascade of PP2Cs-SnRK2s-RAV1/ABF2-ORE1, the ABA receptor PYL9 promotes drought resistance by reducing transpirational water loss and triggering dormancy-like responses such as senescence in old leaves and growth inhibition in young tissues under severe drought conditions [45]. The accelerated leaf senescence observed in transgenic plants overexpressing *PYL9* (under the control of the *pRD29A* promoter) aids in generating a greater osmotic potential gradient, thereby allowing water to preferentially flow to developing tissues [45].

Nonetheless, when exposed to severe drought conditions, the expression of *NTL4* and the accumulation of ROS extend throughout the entire plant, resulting in necrosis of the entire plant body [28]. This observation suggests that delaying leaf senescence could potentially enhance drought tolerance. Under drought stress, maintaining a balance between growth and survival is crucial for the overall fitness of plants [46], yet the mechanisms underlying this balance remain poorly understood [8]. Gaining a deeper understanding of the molecular mechanisms involved in drought-induced leaf senescence holds promise for developing strategies to alleviate the detrimental effects of drought stress on plant growth and productivity [18]. In this regard, NTL4 emerges as a potential candidate gene for coordinating plant stress tolerance and growth by precisely regulating its gene expression to initiate leaf senescence at the appropriate time.

### 2.3. Salt Stress-Induced Leaf Senescence

Salinity, a significant environmental stressor, particularly in arid and semi-arid regions, poses a substantial threat to crop productivity, leading to significant crop losses [47,48]. Salt stress exerts its negative impact on crop growth through various mechanisms, including osmotic stress, toxicity from specific ions, nutrient imbalances, and disrupted hormonal regulation [49,50,51]. It is estimated that more than 6% of the Earth’s land is affected by salinity, with approximately 20% of irrigated land being saline, resulting in substantial agricultural losses amounting to tens of billions of dollars annually [52,53,54]. The effect of salt stress on plant senescence varies depending on the salt concentration. Mild salt stress can induce early flowering in plants, while severe salt stress can trigger leaf senescence and cell death [17].

Several research studies focused on identifying transcription factors involved in the regulation of salt stress-induced leaf senescence [55,56]. One prominent family in this context is the NAC transcription factor family, which has been extensively studied for its role in salt stress-induced leaf senescence [55]. ANAC092/ORE1, a member of this family, was found to contribute to salt-promoted senescence by controlling gene expression in response to salt stress [57]. Overexpressing *ORE1* leads to salt-induced senescence, while *ANAC092* knockout plants exhibit delayed senescence [57]. Ethylene-insensitive 3 (EIN3), a key transcription factor in the ethylene signaling pathway, acts as an upstream regulator of ORE1, influencing both leaf senescence and the response to salt stress [14,58]. Consequently, the age-dependent trigeminal feed-forward pathway involving ANAC092/ORE1 potentially intersects with other developmental and environmental signals to govern leaf senescence and cell death processes [56].

ANAC016 and ANAC032 are additional transcription factors that contribute to the positive regulation of leaf senescence under salt stress by controlling the expression of *SAG*s [59,60,61]. Mutants of *nac016* were found to retain their green phenotype under salt stress conditions, while plants overexpressing *NAC016* exhibit rapid senescence [59]. Similarly, the expression of *ANAC032* was induced by salinity and promotes leaf senescence in response to salt stress [61]. Notably, the *ANAC032OX* line showed increased accumulation of hydrogen peroxide (H_2_O_2_), whereas the chimeric repressor line (*ANAC032-SRDX*) exhibited reduced H_2_O_2_ levels [61]. These findings suggest that the altered responses of ANAC032 transgenic lines to salt stress may involve differential accumulation of ROS [61]. ANAC047, another transcription factor induced by salinity, is also implicated in salt stress-induced senescence [62]. Transgenic plants expressing the chimeric inhibitor *ANAC047-SRDX* displayed enhanced salt tolerance, indicating that ANAC047 acts as a positive regulator of stress-induced senescence [62]. Conversely, ANAC083/VNI2 functions as a negative regulator of senescence in Arabidopsis [63]. Plants with high expression levels of *ANAC083* exhibited significant salt and drought tolerance, along with delayed senescence [63]. Moreover, increased *ANAC083* expression led to the upregulation of *COR/RD* genes [63]. ANAC042/JUNGBRUNNEN1 (JUB1), another negative regulator of senescence, promotes plant longevity and confers tolerance to abiotic stresses such as heat and salt in Arabidopsis [64]. *JUB1* expression is rapidly induced by the accumulation of H_2_O_2_, and its overexpression results in delayed natural senescence [64]. Recently, a transcription factor from the AP2/ERF family, ethylene-responsive factor 34 (ERF34), was identified as a negative regulator of salt stress-induced leaf senescence and a contributor to salt stress tolerance [56]. ERF34 directly binds to the promoters of early responsive to dehydration 10 (*ERD10*) and responsive to desiccation 29A (*RD29A*), activating their expression [56]. This study suggests that ERF34 may serve as a potential mediator that integrates salt stress signals with the leaf senescence program.

The pivotal role of stress response transcription factors as key regulators of leaf senescence was extensively demonstrated in crops and trees [65,66,67]. For instance, overexpression of the rice NAC gene *SNAC1* in transgenic cotton enhances drought and salt tolerance by promoting root development and reducing transpiration rate [68]. In rice, the salt stress response gene ONAC106 acts as a negative regulator of leaf senescence [69]. Gain-of-function mutants of *ONAC106*, such as *ONAC106-1D* transgenic plants with a 35S enhancer inserted into the *ONAC106* gene’s promoter region, exhibited delayed senescence and improved salt stress tolerance [69]. Similarly, the overexpression of *ShNAC1* in *Solanum habrochaites* delays salt stress-induced leaf senescence [70]. In *Populus euphratica*, the overexpression of two NAC transcription factors, *PeNAC034* and *PeNAC036*, results in enhanced salt stress sensitivity and tolerance, respectively [71]. Notably, *PeNAC034* overexpression promotes leaf senescence, while *PeNAC036* overexpression inhibits it [72]. In addition to transcription factors, other regulatory genes also play a crucial role in salt-induced leaf senescence. In rice, the loss of function of the receptor-like kinase gene bilateral blade senescence 1 accelerates leaf senescence and reduces salt tolerance [73].

The overexpression of the salt-inducible protein *salT* in rice was shown to delay leaf senescence, potentially serving as a feedback regulation to suppress salt stress-induced senescence [74]. Furthermore, a comparative transcriptome analysis of Arabidopsis plants exposed to age-dependent and salt stress-induced leaf senescence revealed potential molecular mechanisms underlying the interplay between these two senescence scenarios, including the involvement of H_2_O_2_-mediated signaling [75]. Salt stress-induced leaf senescence is a complex process regulated by multiple genes and signaling pathways. However, the intricate mechanisms that integrate salt stress signaling with the leaf senescence program remain largely elusive [56]. Enhancing our understanding of the molecular mechanisms underlying salt-induced leaf senescence will contribute to the development of strategies aimed at improving plant stress tolerance and crop productivity [8].

### 2.4. Darkness-Induced Leaf Senescence

Light plays a crucial role in plant growth, morphology, and development [76]. However, when plants are exposed to shade or complete darkness for an extended period, it triggers leaf senescence [37,77,78,79,80,81]. Transcriptomic analysis has shown that gene expression changes induced by darkness closely resemble those observed during natural senescence [82,83,84,85]. In fact, more than 50% of the genes up-regulated during natural senescence are also up-regulated under dark treatment conditions [83]. As a result, dark treatments are widely employed as a rapid, convenient, and effective method to induce leaf senescence, making it easier to investigate the impact of additional regulators of senescence, such as phytohormones, sugars, and secondary metabolites [8,83].

Recent investigations unveiled several genes and signaling pathways associated with dark-induced leaf senescence. To identify mutants with delayed dark-induced senescence, an experiment utilizing an individually darkened leaf (IDL) setup was conducted on *Arabidopsis thaliana* Col-0 plants treated with ethyl methanesulfonate mutagenesis [80]. The study revealed that PIF5 loss-of-function mutants, specifically *pif5-621*, exhibited significantly delayed chlorophyll loss in the IDL [80]. Remarkably, the overall growth habit of *pif5-621* resembled that of wild-type plants, indicating a direct impact of the *pif5* mutation on senescence rather than an indirect effect through life cycle progression or overall growth [80]. One plausible hypothesis to explain the extended lifespan of *pif5-621* IDLs is that the cells decelerated their metabolism, particularly respiration, to minimize carbon consumption and prolong survival compared to wild-type IDLs. Supporting this notion, *pif* quadruple mutants (*pifQ*) *pif1 pif3 pif4 pif5*, which exhibit a constitutive photomorphogenic phenotype when grown in the dark, maintained green cotyledons even after 10 days of dark treatment, while cotyledons of the wild type turned completely yellow, indicating that PIFs promote senescence under light-deprived conditions [38,40]. PIF4 and PIF5 influence ABA signaling by modulating ABSCISIC ACID INSENSITIVE 5 (ABI5) and ENHANCED EM LEVELS (EEL), two sister genes encoding basic leucine zipper (bZIP) class A transcription factors, which exhibited significantly reduced induction after darkening in *pifQ* mutants compared to the wild type [40]. Correspondingly, the single mutants *abi5*, *eel*, and, particularly, the *abi5 eel* double mutant displayed delayed senescence under dark conditions. Furthermore, PIF4 or PIF5 stimulates ethylene signaling by directly regulating the transcription of *EIN3* [40]. Additionally, ethylene evolution is diminished in *pif4* mutants and elevated in *PIF4* and *PIF5* overexpressors [38,86]. Treatment of *pif4* mutants with ethylene partially restored the senescence phenotype, indicating that PIFs promote dark-induced senescence by inducing ethylene biosynthesis and signaling. Moreover, PIF4, PIF5, and their target transcription factors (ABI5, EEL, and EIN3) directly activate the transcription of *ORE1*, suggesting the establishment of multiple coherent feed-forward regulatory circuits involving these transcription factors to induce leaf senescence [37]. As expected, *ein3* and *ore1* mutants exhibited a significant delay in senescence compared to the wild type, as evidenced by higher chlorophyll content and Fv/Fm levels under dark conditions [14]. PIF4/PIF5 directly activates the expression of *ABI5* and *EIN3*, which, in turn, activate the transcription of *ORE1*. ORE1 collaborates with PIFs, ABI5, and EIN3 to up-regulate genes involved in chlorophyll degradation, including staygreen 1 (*SGR*) and non-yellow coloring 1 (*NYC1*) [38,40,87]. Conversely, ORE1 interacts with PIFs to suppress the chloroplast maintenance master regulators GOLDEN2-LIKE 1 (GLK1) and GLK2. This antagonistic action of ORE1 on GLKs shifts the balance from chloroplast maintenance to deterioration [88].

Apart from the ABA and ethylene signaling pathways, dark-induced leaf senescence also involves the participation of JA. The genes responsible for JA biosynthesis, namely lipoxygensase 2 (*LOX2*) and allene oxide synthase (*AOS*), are up-regulated during dark-induced leaf senescence, and the application of exogenous JA expedites the senescence process [89]. Overall, the induction of leaf senescence by dark treatment is governed by an intricate network of molecular mechanisms encompassing various genes and regulatory pathways. The identification of these pivotal genes and pathways offers valuable insights into the regulatory mechanisms underlying dark-induced leaf senescence and holds potential for the development of strategies aimed at delaying or preventing leaf senescence in crop plants.

### 2.5. Low Oxygen-Induced Leaf Senescence

Low oxygen, also referred to as hypoxia, represents an abiotic stress condition capable of triggering leaf senescence in plants [90,91,92]. In response to low oxygen levels, plants activate various adaptive mechanisms to maintain cellular homeostasis and minimize oxidative damage. However, prolonged exposure to hypoxia can accelerate leaf senescence, leading to reduced plant growth and yield. Notably, leaf senescence is a prominent visible symptom observed in plants subjected to extended submergence [90,91,92]. Chlorophyll degradation initiates during the hypoxic phase and becomes evident after prolonged submergence (typically lasting 5 to 7 days) in rice and Arabidopsis [90,91,92].

At the molecular level, the regulation of hypoxia-induced leaf senescence involves a complex interplay of genes and signaling pathways. Among the key contributors to this process are the transcription factors belonging to the group VII ethylene response factor (ERFVIIs), which stabilize under hypoxic conditions and activate downstream gene expression to facilitate plant adaptation to low oxygen levels [93,94,95]. In rice, the ERFVII transcription factor known as submergence 1a (SUB1A) functions as a regulator of submergence tolerance by attenuating leaf senescence during prolonged submergence. Through functional characterization, it was revealed that the induction of *SUB1A* expression during submergence restricts further ethylene production and reduces gibberellic acid responsiveness. As a result, shoot tissues experience a decrease in carbohydrate consumption, chlorophyll breakdown, amino acid accumulation, and elongation growth [90,91,92]. This quiescence response to submergence aids in preserving carbohydrate reserves and the capacity for photosynthesis. The prevention of carbohydrate depletion may contribute to the milder manifestation of leaf senescence observed during submergence [96].

Interestingly, ectopic overexpression of *SUB1A* not only delays darkness-induced leaf senescence but also limits ethylene production and responsiveness to JA and salicylic acid (SA). This suppression of ethylene, JA, and SA signaling pathways results in the preservation of chlorophyll and carbohydrates [97]. The delay in leaf senescence conferred by SUB1A contributes to enhanced tolerance to submergence, drought, and oxidative stress [96,97,98]. Collectively, the molecular mechanisms governing hypoxia-induced leaf senescence are intricate and multifaceted. Gaining a comprehensive understanding of these mechanisms is crucial for the development of strategies aimed at enhancing plant tolerance to hypoxia and mitigating its adverse effects on plant growth and yield.

### 2.6. Extreme Temperatures Stress-Induced Leaf Senescence

Heat stress is one of the major environmental factors that trigger precocious senescence in plans. Heat-stress-induced leaf senescence is associated with ethylene accumulation and chlorophyll loss [2,99]. High-temperature treatment increased ethylene production in soybean (*Glycine max*) leaves and pods, which may be due to higher ACC synthase activity [99]. Pheophytinase (PPH) could be one of enzymes that play key roles in regulating heat-accelerated chlorophyll degradation [100]. After heat stress, the survival rate of *pph* mutant plants was significantly higher than that of wild type plants. It also led to a significant decrease in chlorophyll content in wild type plants and *pph* mutants, but the decrease was greater in wild type plants. The previously mentioned PIF4 and PIF5 are key regulators of heat-induced senescence [37,38,39,40]. Under heat stress, leaf senescence was delayed in *pif4* and *pif5* mutants and accelerated in transgenic lines compared with the wild type. *NAC019*, *SAG113*, and *IAA29* were characterized as direct targets of PIF4 and PIF5. In addition, PIF4 and PIF5 proteins accumulate with the progression of heat stress-induced leaf senescence and are regulate at the transcriptional and posttranscriptional levels [101]. In addition, mutation of premature senescence leaf 50 (PSL50) led to higher heat sensitivity, reduced survival, excessive hydrogen peroxide (H_2_O_2_) content, and increased cell death under heat stress in rice. This result suggests that PSL50 improves heat tolerance by regulating H_2_O_2_ signaling under heat stress [102]. Low temperature and short day length could result in the decrease in cytokinin and the increase in abscisic acid in leaf tissue, which directly trigger/promote senescence [103], which was supported by another study [104]. So far, low-temperature-induced leaf senescence has not been well-studied, and the underlying molecular regulatory mechanisms remain to be explored.

### 2.7. Other Abiotic Stresses-Induced Leaf Senescence

Apart from the previously mentioned abiotic stresses, additional factors such as extreme temperatures, high sugar levels, and UV radiation can also trigger premature leaf senescence [2,8]. Elevated sugar levels within plant tissues lead to reduced photosynthesis and early onset of senescence. The loss of hexokinase-1 (HXK1) function results in a delayed senescence phenotype [105], whereas the overexpression of *Arabidopsis HXK1* (*AtHXK1*) in tomato plants accelerates senescence [106]. These findings indicate the involvement of the sugar sensor HXK1 in sugar signaling during senescence. Intriguingly, the *hxk/gin2* mutant does not accumulate hexose in senescing leaves [107]. Moreover, the *hxk/gin2* mutant exhibits a delay in senescence induction by externally supplied glucose [105], suggesting that HXK1 plays a role in sugar metabolism and response during senescence. Notably, growth on glucose in combination with low nitrogen supply induces leaf yellowing and alters gene expression patterns, characteristic of developmental senescence. Importantly, the senescence-specific gene *SAG12* is significantly upregulated by glucose. Additionally, two senescence-associated MYB transcription factor genes, production of anthocyanin pigment 1 (*PAP1*) and *PAP2*, are induced by glucose [108]. In Arabidopsis, glucose and fructose accumulate substantially during leaf developmental senescence, while the sucrose content remains relatively unchanged [107]. Generally, the sugar content in leaves gradually increases, reaching its peak during the mature green stage or early senescence stages. Although the mechanisms underlying the maintenance of carbon storage molecules, such as sugars and starch, during senescence are not fully understood, sugars undoubtedly play a crucial role in driving cellular processes in senescing leaves [109].

Moreover, the presence of heavy metal pollutants, such as cadmium, poses a significant environmental challenge, leading to detrimental effects on plant growth and development. Cadmium toxicity triggers the generation of ROS, disrupts the photosynthetic system, and disrupts nutrient balance, ultimately accelerating leaf senescence [110,111]. Intriguingly, the accumulation of cadmium in leaves increases exponentially during the senescence process [112], indicating a clear association between leaf senescence and cadmium accumulation. However, the exact mechanism of cadmium accumulation in senescing leaves and the causal relationship between cadmium accumulation and senescence remain unclear. In particular, senescing leaves of tall fescue (*Festuca arundinacea*) can serve as a means to remove cadmium from polluted soil through a sustainable approach known as phytoextraction [110,112].

## 3. Improvement of Stress Tolerance through Regulation of Leaf Senescence

Understanding leaf senescence holds great importance due to its potential for improving crop yield and quality. Manipulating the timing of leaf senescence enables plant breeders to enhance photosynthetic efficiency, nutrient absorption, and stress tolerance, leading to increased crop yield and improved quality. Additionally, leaf senescence plays a pivotal role in plant adaptation to environmental stress. Exploring leaf senescence provides valuable insights for developing stress-tolerant plants capable of withstanding adverse conditions like drought, heat, or cold, thereby minimizing the detrimental effects of these stressors on plant growth and productivity.

### 3.1. Utilization of Senescence-Specific or Stress-Associated Promoters

By utilizing the promoter of a senescence-specific gene *SAG12*, Gan and Amasino designed an ingenious and elegant auto-regulatory senescence-inhibition system, *pSAG12-IPT* [25] (Figure 1A). The promoter of *SAG12* was linked to the coding region of the isopentenyltransferase gene (IPT), which regulates the rate-limiting step in cytokinin biosynthesis, to form the chimeric gene *pSAG12-IPT* [25]. At the onset of senescence, this promoter activates IPT expression and increases cytokinin content to levels that prevent leaf senescence. Repression of senescence in turn attenuates promoter expression to prevent overproduction of cytokinin. The use of senescence promoters is essential to avoid premature IPT overexpression and CK hyper-production ahead of senescence. The auto-regulatory biosynthetic system using p*SAG12-IPT* was proven to be an effective strategy for developing transgenic plants to increase yield by delaying senescence and extending the shelf life of isolated organs such as leaves, flowers, and fruits [28]. The *pSAG12-IPT* system had been widely used in numerous plant species [28], including wheat (*Triticum aestivum* L.) [113], alfalfa (*Medicago sativa*) [114], lettuce (*Lactuca sativa* L. *cv Evola*) [115], cassava (*Manihot esculenta Crantz*) [27], and creeping bentgrass (*Agrostis stolonifera* L. *‘Penncross’*) [116], etc. However, it should be noted that the pSAG12-IPT system possibly directly or indirectly affects plant development, including delayed flowering in transgenic lettuce [115], and reduced nitrogen accumulation in young leaves by altering sink-source relationships in tobacco [117]. To achieve maximum effectiveness, practical applications should carefully consider the advantages and disadvantages of this system.

A range of variants were developed based on the design concept of *pSAG12-IPT*. One approach involves utilizing different promoters to control the expression of *IPT*. For instance, a modified version of this cytokinin (CK) auto-regulatory cycle strategy employed the promoter of *senescence-associated receptor kinase* (*SARK*) fused with the *IPT* gene (Figure 1B). Transgenic tobacco plants carrying *pSARK-IPT* exhibited enhanced survival under severe drought conditions, accompanied by improvements in photosynthetic rate and water use efficiency [26]. In these plants, the activation of the *SARK* promoter in response to drought-induced leaf senescence led to delayed senescence through cytokinin biosynthesis. However, it should be noted that premature activation of leaf senescence may occur during drought, and the benefits in terms of yield increase may not be realized under well-watered conditions when using stress-inducible promoters. To address the issues of stress inducibility and proper regulation of *IPT* genes, Spangenberg and colleagues ingeniously employed a modified promoter derived from the developmental process-related gene *AtMYB32* (*AtMYB32xs*) (Figure 1C), which removed the 360 bp root-specific motif [118]. Stable transgenic oilseed rape (*Brassica napus*) plants expressing *AtMYB32xs-IPT* exhibited delayed leaf senescence under controlled environment and field conditions. Remarkably, these *AtMYB32xs-IPT* plants achieved significantly higher seed yield during both rainy seasons and field irrigation conditions [118]. In petunia and chrysanthemum, transgenic plants known as *COR15A-IPT* were generated using the cold induction promoter from the cold-regulated15a (*COR15A*) gene of *Arabidopsis thaliana* (Figure 1D) [119]. Intriguingly, *COR15A-IPT* plants and their detached leaves remained green and healthy during extended dark storage (4 weeks at 25 °C) following an initial exposure to a brief period of cold induction (72 h at 4 °C). This study presented an approach to prolong the lifespan of transplants or excised leaves during storage under dark and cold conditions, which is particularly beneficial for long-distance transport. The heat shock promoter *HSP18.2* was fused with *IPT* to generate *HSP18.2-IPT* transgenic plants in creeping bentgrass (*Agrostis stolonifera*) (Figure 1E) [116]. The *HSP18.2-IPT* transgenic lines exhibited significantly improved turf quality, photochemical efficiency, chlorophyll content, relative leaf water content, and root-to-stem ratio. Furthermore, transgenic poplar lines expressing *IPT* under the control of the promoter of *PtRD26* (*PtRD26_pro_-IPT*) (Figure 1F), a senescence and drought-inducible NAC transcription factor in poplar, displayed various phenotypic improvements, including enhanced growth and drought tolerance [120].

Another type of experimental design is to use promoters of *SAG12* to drive the expression of different genes. For example, tobacco plants overexpressing the maize homeobox gene *knotted1* (*kn1*) under the driver of *SAG12* promoter, designated as *pSAG12-kn1*, exhibited a significant delay in leaf senescence, with an increase in chlorophyll content and a decrease in the number of dead leaves. In the detached leaves of *pSAG12-kn1* plants, senescence was also postponed [121]. Collectively, these studies provided the possibility of regulating the onset of leaf senescence by cleverly using senescence and stress-related gene promoters to drive the expression of *IPT* and developmental genes, thereby improving crop resistance, yield, and quality.

### 3.2. Modulation of Expression of Senescence Associated Genes

An alternative approach to influencing the leaf senescence process involves manipulating the expression of crucial senescence genes, with the aim of enhancing crop resistance and yield. For instance, the knockout of *OsNAP*, a rice ortholog of *ANAC029/AtNAP* [122], resulted in prolonged grain-filling periods and increased grain yields compared to the wild type [123]. Therefore, precise regulation of *OsNAP* expression holds promise for improving stress resistance in rice. A noteworthy discovery is the potential use of naturally occurring *Stay-Green* (*OsSGR*) promoter and associated longevity variants in breeding programs to enhance rice yield [124]. Nam and colleagues conducted quantitative trait loci (QTL) mapping and identified genetic differences in life cycle and senescence patterns between two rice subspecies, indica, and japonica [124]. They found that promoter variations in the *OsSGR* gene, which encodes the chlorophyll-degrading Mg^++^-dechelatase, triggered earlier and higher induction of *OsSGR* in indica, thereby accelerating senescence in indica cultivars. Introducing the japonica OsSGR allele into indica-type cultivars resulted in delayed senescence, increased grain yield, and improved photosynthetic capacity. This study highlighted the potential of modifying the senescence-related promoter region, in addition to gene coding regions, to achieve delayed leaf senescence and increased yield. The use of gene editing technologies like CRISPR/Cas9 offers a powerful tool for manipulating key genes involved in regulating leaf senescence, and further research is necessary to identify and manipulate additional genes involved in these processes [8].

To summarize, comprehending the molecular regulatory mechanisms underlying leaf senescence holds the potential to inform the development of molecular breeding strategies aimed at enhancing plant tolerance to abiotic stresses, increasing grain yield, and improving crop quality. A significant objective in leaf senescence research is the cultivation of plants with ideal leaf senescence phenotype (PILSP). PILSP exhibit the remarkable ability to effectively coordinate growth and stress tolerance, integrate both internal and external signals, and initiate leaf senescence at the optimal time. In the case of annual plants, leaves remain green in the initial stages of plant growth, resisting internal and external stresses, and only enter senescence when the leaves die, allowing for the complete transfer of photosynthetic products to the seeds. In contrast, perennial plants retain their leaves even under environmental stresses, and upon stress removal, leaf function is promptly restored. Consequently, when the leaves eventually die, the photosynthetic products can be fully channeled to the main stem or the growing organ.

## 4. Prospects

In recent years, considerable advancements have been made in leaf senescence research; however, several challenges remain in fully comprehending the molecular mechanisms of leaf senescence and effectively applying this knowledge to enhance crop improvement. One of the foremost challenges lies in the intricate nature of the regulatory network governing leaf senescence, posing difficulties in identifying pivotal genes and regulatory pathways. Additionally, the complexity is compounded by the fact that diverse environmental factors, such as drought, high temperature, and nutrient deficiency, can trigger leaf senescence through distinct pathways, further adding to the intricacy of the regulatory network.

To overcome these challenges, several strategies have been proposed. Firstly, a comprehensive analysis of the leaf senescence regulatory network and the identification of key genes and regulatory pathways can be achieved through the integration of multi-omics approaches, including genomics, transcriptomics, proteomics, and metabolomics [19,125,126,127]. These approaches enable a holistic understanding of the complex mechanisms involved. Secondly, advanced imaging techniques, such as live imaging and high-resolution microscopy, offer the opportunity to monitor the dynamic progression of leaf senescence and visualize the molecular events at play. For instance, confocal imaging fluorometer allows high spatio-temporal-resolution detection of chlorophyll fluorescence dynamics at the single chloroplast level [128]. Additionally, the combination of high-speed three-dimensional laser scanning confocal microscopy and high-sensitivity multiple-channel detection facilitates in-depth investigations of the spatial and temporal dynamics of chloroplast degradation during leaf senescence [129]. Thirdly, genetic engineering techniques, particularly CRISPR/Cas9-mediated genome editing, provide a means to manipulate the expression of key senescence-related genes and elucidate their roles in the process. A recent study successfully employed CRISPR/Cas9-mediated knockout to demonstrate the regulatory function of the peptide hormone CLE42 in leaf senescence [130].

In conclusion, leaf senescence research holds immense potential for enhancing crop yield and quality. However, addressing the existing challenges is crucial. By harnessing the power of multi-omics approaches, advanced imaging techniques, and genetic engineering, we can gain a deeper understanding of the molecular mechanisms underlying leaf senescence and effectively apply this knowledge to drive crop improvement.

## Figures and Tables

**Figure 1 ijms-24-11996-f001:**
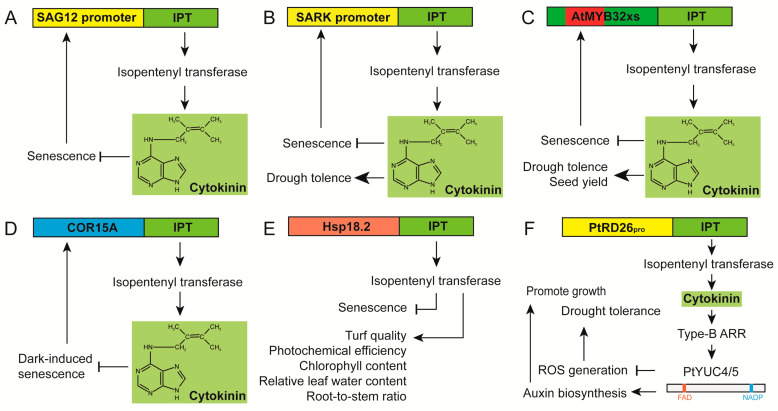
The diagrams show *pSAG12-IPT* and its various variants. (**A**) The auto-regulatory senescence-inhibition system of *pSAG12-IPT* [25]. (**B**–**F**) A range of variants have been developed based on the design concept of *pSAG12-IPT*, including (**B**) *pSARK-IPT* [26], (**C**) *AtMYB32xs-IPT* (Red rectangle represent the deleted root motif of 360bp in promoter) [118], (**D**) *COR15A-IPT* [119], (**E**) H*SP18.2-IPT* [116], and (**F**) *PtRD26_pro_-IPT* [120]. The yellow parts represent the senescence-specific promoters. The dark green part represents the developmental process-related promoter. The blue part represents the cold-induced promoter. The red part represents the heat shock promoter.

## Data Availability

No new data were created or analyzed in this study. Data sharing is not applicable to this article.

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
