# Peer review of "Abiotic Stress-Induced Leaf Senescence: Regulatory Mechanisms and Application"

_ijms, 2023, doi:10.3390/ijms241511996_

Round 1

Reviewer 1 Report

This article treats plants as a whole; there is no distinction between annuals and perennials, monocots and dicots. However, an outline of some universal cascade of physiological and cellular events during the senescence process would be nice. What, apart from activating the appropriate genes, happens physically during the aging of the leaves?

The review may be more interesting if it highlights any specific adaptations or strategies employed by annual and perennial plants to cope with abiotic stresses and delay leaf senescence. And examine the role of the plant's life cycle and the timing of stress exposure in shaping the senescence responses.

There needs to be more information in this article regarding aging caused by low or high temperatures. Due to the climate crisis, there are often days with non-standard temperatures for a given area. Reading a separate section on temperature-induced aging would help Readers to look at this process from a new perspective.

Author Response

This article treats plants as a whole; there is no distinction between annuals and perennials, monocots and dicots. However, an outline of some universal cascade of physiological and cellular events during the senescence process would be nice. What, apart from activating the appropriate genes, happens physically during the aging of the leaves?

The review may be more interesting if it highlights any specific adaptations or strategies employed by annual and perennial plants to cope with abiotic stresses and delay leaf senescence. And examine the role of the plant's life cycle and the timing of stress exposure in shaping the senescence responses.

Response: This is a very valuable suggestion. However, our manuscript was not reviewed by classification of monocots, dicots, annual and perennial. If so, the organization of the article will change a lot. We plan to follow your suggestions in a future paper.

There needs to be more information in this article regarding aging caused by low or high temperatures. Due to the climate crisis, there are often days with non-standard temperatures for a given area. Reading a separate section on temperature-induced aging would help Readers to look at this process from a new perspective.

Response: Many thanks for your constructive comments. As requested, we had added a section about extreme temperature-induced leaf senescence.

Reviewer 2 Report

This paper is a wide contribution to the unveiling of abiotic stress induced leaf senescence in plants. Nevertheless, some references related to the modulation of expression of senescence associated genes are missing,  for example: Moschen S., et al  2014 “Identification of candidate genes associated with leaf senescence in cultivated sunflower (Helianthus annuus L.)” (2014). PLoS ONE 9(8): e104379. doi:10.1371/journal.pone.0104379;  “Integration of transcriptomic and metabolic data reveals hub transcription factors involved in drought stress response in sunflower (Helianthus annuus L.)”. Moschen et al (2017). Plant Molecular Biology DOI: 10.1007/s11103-017-0625-5; “Identification and expression analysis of NAC transcription factors potentially involved in leaf and petal senescence in Petunia hybrida” (Trupkin et al 2019).

Some minor edition on format and sentences should be revised and edited.

Author Response

The authors have submitted a well-written review on abiotic stress-induced leaf senescence. The strengths of the article are the clear explanations of published work, not a long list with little narrative. I found the review to be readable and helpful in identifying a few studies I had missed.

Response: We highly appreciate your positive comments.

For the first two sections, two small comments:

Lines 236-238: The effects of PeNAC034 and PeNAC036 overexpression are reversed in the review. In the original paper, PeNAC034 enhanced salt/drought sensitivity while PeNAC036 increased salt/drought tolerance.

Response: Many thanks for pointing this out. As requested, we had revised the manuscript.

Lines 375-376: The increase in cadmium was observed in tall fescue and should not be generalized for other plant species.

Response: As requested, we had revised this manuscript.

For the third section, there are more problems to address:

Line 407: The developmental problems associated with pSAG12-IPT need to be explained. These include delayed flowering (PMCID: PMC125086) and altered sink-source dynamics (doi: 10.1007/s00425-005-1489-5 and https://doi.org/10.1046/j.1365-3040.2000.00544.x). Other effects are inhibited root growth and increased development of axillary buds.

Response: As requested, we added a few sentences to discuss these problems with this system.

Line 439(495): The AtMYB32xs promoter had a 360 bp deletion and multiple motifs were deleted. The original paper does not clearly show the new expression pattern. A more detailed and critical view of this paper is needed.

Response: Many thanks for pointing this out. As requested, we had made the corresponding revision in Figure 1.

The review does not mention two important papers where leaf senescence is early, and yields are increased. One studies the wheat NAM genes (doi: 10.1093/g3journal/jkac275) and the other focuses on overexpression of OsDREB1C (DOI: 10.1126/science.abi8455). These are just two examples of such findings which must be included in this review.

Response: Many thanks for your constructive comments. As requested, we had added these descriptions.

In addition, lines 408-424 repeat 392-407.

Response: Many thanks for pointing this out. As requested, we have deleted the duplicate paragraphs.

Reviewer 3 Report

The authors have submitted a well-written review on abiotic stress-induced leaf senescence. The strengths of the article are the clear explanations of published work, not a long list with little narrative. I found the review to be readable and helpful in identifying a few studies I had missed.

For the first two sections, two small comments:

Lines 236-238: The effects of PeNAC034 and PeNAC036 overexpression are reversed in the review. In the original paper, PeNAC034 enhanced salt/drought sensitivity while PeNAC036 increased salt/drought tolerance.

Lines 375-376: The increase in cadmium was observed in tall fescue and should not be generalized for other plant species.

For the third section, there are more problems to address:

Line 407: The developmental problems associated with pSAG12-IPT need to be explained. These include delayed flowering (PMCID: PMC125086) and altered sink-source dynamics (doi: 10.1007/s00425-005-1489-5 and https://doi.org/10.1046/j.1365-3040.2000.00544.x). Other effects are inhibited root growth and increased development of axillary buds.

Line 439: The AtMYB32xs promoter had a 360 bp deletion and multiple motifs were deleted. The original paper does not clearly show the new expression pattern. A more detailed and critical view of this paper is needed.

The review does not mention two important papers where leaf senescence is early, and yields are increased. One studies the wheat NAM genes (doi: 10.1093/g3journal/jkac275) and the other focuses on overexpression of OsDREB1C (DOI: 10.1126/science.abi8455). These are just two examples of such findings which must be included in this review.

In addition, lines 408-424 repeat 392-407.

Author Response

Comments and Suggestions for Authors

This paper is a wide contribution to the unveiling of abiotic stress induced leaf senescence in plants. Nevertheless, some references related to the modulation of expression of senescence associated genes are missing,  for example: Moschen S., et al  2014 “Identification of candidate genes associated with leaf senescence in cultivated sunflower (Helianthus annuus L.)” (2014). PLoS ONE 9(8): e104379. doi:10.1371/journal.pone.0104379;  “Integration of transcriptomic and metabolic data reveals hub transcription factors involved in drought stress response in sunflower (Helianthus annuus L.)”. Moschen et al (2017). Plant Molecular Biology DOI: 10.1007/s11103-017-0625-5; “Identification and expression analysis of NAC transcription factors potentially involved in leaf and petal senescence in Petunia hybrida” (Trupkin et al 2019).

Response: Many thanks for your constructive comments. As requested, we had added some descriptions about the regulation of leaf senescence and cited these references.

Comments on the Quality of English Language

Some minor edition on format and sentences should be revised and edited.

Response: As requested, we had revised the manuscript thoroughly.